# Sedentary Behaviour and Physical Activity Levels during Second Period of Lockdown in Chilean’s Schoolchildren: How Bad Is It?

**DOI:** 10.3390/children10030481

**Published:** 2023-03-01

**Authors:** Ricardo Martínez-Flores, Ignacio Castillo Cañete, Vicente Pérez Marholz, Valentina Marín Trincado, Carolina Fernández Guzmán, Rodrigo Fuentes Figueroa, Gabriela Carrasco Mieres, Maximiliano González Rodríguez, Fernando Rodriguez-Rodriguez

**Affiliations:** 1IRyS Group, School of Physical Education, Pontificia Universidad Católica de Valparaíso, Viña del Mar 2340000, Chile; 2School of Physical Education, Pontificia Universidad Católica de Valparaíso, Viña del Mar 2340000, Chile

**Keywords:** screen time, sedentary behaviour, quarantine, isolation, coronavirus

## Abstract

Objective. The objective of this study was to compare the levels of sedentary behaviour and physical activity in relation to sociodemographic variables of Chilean schoolchildren before and during the COVID-19 pandemic. Methods. This retrospective study considered a non-random sample of 83 boys and 232 girls, and their respective parents, who attended public schools (*n* = 119) and private schools (*n* = 196) in Chile. A self-report instrument was applied that included sociodemographic variables, sedentary behaviour (SB), and physical activity (PA) in the second period of the pandemic in 2021. Results. The main results show that pre-pandemic SB had significant differences when compared between sexes, except for television time. During the pandemic, there was no significant difference in television time or telephone time. There were no significant differences by sex before and during the pandemic. When comparing the SB scores, video game time in boys decreased (*p* < 0.001), as did video game time in girls (*p* < 0.001), and computer time in boys (*p* < 0.001) and girls (*p* < 0.001). Telephone time increased in boys (*p* < 0.001) and girls (*p* < 0.001), as did television time (*p* < 0.001). Likewise, PA increased in boys (Δ + 9.51min) and girls (Δ + 3.54 min) during the pandemic (*p* < 0.001). Conclusions. Both PA and SB underwent changes according to sex before and during the second period of the COVID-19 pandemic in Chilean schoolchildren.

## 1. Introduction

The global population has been severely affected by the COVID-19 pandemic: more than 4.6 million people have been affected by the disease, including children and young people [1]. As a preventive measure, millions of people worldwide have practised social distancing, isolation, and quarantine. However, compliance with these measures has brought with it health problems related to decreased physical activity (PA) and increased sedentary behaviour (SB), as well as a psychological impact associated with the pandemic’s state of uncertainty [2]. In Chile, multiple dimensions of daily life have also been affected, such as the development of social and educational relationships [3], due to mobility restrictions and confinement, causing children and youth to spend a lot of time at home [4].

Current literature points out that the lifestyle caused by COVID-19 has led to an increased prevalence of SB [5,6], which is defined as behaviours that do not exceed a basal energy expenditure of 1.5 METs—metabolic equivalent of task—[7], such as lying down or sitting in front of a screen, among other behaviours [8]. Additionally, being physically inactive corresponds to not complying with international PA recommendations [9]. For 15 years, the global pattern of physical activity did not present significant modifications, highlighting Latin America as having a higher prevalence of physical inactivity [10]. Regarding PA during the pandemic, recent studies have found a decrease in PA in both children and adolescents [11], and the older the child, the lower the level of PA [12]. Furthermore, there was a decrease in PA during the pandemic in children aged 1 to 5 years old in Chile [13]. Additionally, data collected on sedentary lifestyles show that Chilean schoolchildren spend more than half of their free time sitting as a consequence of confinement, intensifying the risk of acquiring different diseases in the future [14]. In addition, the amount of screen time spent by Chilean children has been shown to be higher than it was before the pandemic [15].

Sociodemographic factors could also be associated with levels of PA [16]. For example, boys have higher levels of PA than girls [17,18] mainly for social reasons. Additionally, Gutiérrez et al. [19] point out that socioeconomic status can be associated with differences in PA levels, where people with high socioeconomic income achieve higher PA levels. Likewise, family characteristics are influential for the active behaviours of children and adolescents, with family members acting as promoters, companions, and facilitators of active behaviours [20]. In this sense, parental behaviours related to physical activity and eating habits are of utmost importance for the same behaviours in schoolchildren [21].

These differences in levels of PA due to the low socioeconomic status of children and their families can impact children’s physical and mental well-being and lead to larger social and educational gaps [22]. Likewise, previous evidence also suggests that PA can be an answer to narrowing these gaps [23,24,25]. However, the influence of the level of PA and SB, according to sociodemographic factors related to family environment, has not been studied in schoolchildren during the period of lockdown during the pandemic.

Consequently, the objective of this study has been to determine the impact of the second period of confinement due to the COVID-19 pandemic on sedentary behaviours and the level of physical activity in Chilean school boys and girls, with the intention of recognizing the real magnitude of the issue. This is to deliver information that allows the authorities to take improvement measures and changes aimed at schoolchildren and their families.

## 2. Materials and Methods

### 2.1. Study Design and Participants

This study has a retrospective design and employs a non-random sample of voluntary participants. A total of 315 schoolchildren (83 boys and 232 girls), and their respective parents, from public (37.8%) and private (62.2%) schools participated. The families were divided into those with only one child and those with more than one child. The availability of a car was also considered a good predictor of socioeconomic level, and those families with no car were separated from those with only one car or more than one car. Finally, we also separated the sample by socioeconomic status: low, medium, and high.

The data were collected between 1 August and 10 September 2021, while most of the schools were still closed. The inclusion criteria were schoolchildren and adolescents between 8 and 14 years of age and their parents or guardians who had voluntarily agreed to participate in the study. The exclusion criteria corresponded to students or parents with visible physical or cognitive difficulties (medically confirmed), which could affect their health by participating in the study. In addition, schoolchildren < 8 years and >14 years and parents or guardians < 18 years were excluded. This study only analysed the results obtained from schoolchildren and does not include information from parents.

### 2.2. Instruments

A questionnaire was a self-reported online survey undertaken in Spanish from July to September 2020 using the SurveyMonkey platform (San Mateo, CA, USA), and was composed of four items that included sociodemographic characteristics (Appendix A). The first section covered sociodemographic characteristics such as sex (male, female), age (in years), residence (urban, rural), number of children at home, and school type (private, public). To establish the socioeconomic level, the level of family wealth was used, through The Family Affluence Scale (FAS) questionnaire, which estimates socioeconomic level according to the number of vehicles, place of residence, type of room, family vacations in the last 12 months, and computers available at home [22]. To determine the PA of schoolchildren, the Youth Activity Profile (YAP) questionnaire was used, which assesses the level of PA in and out of school. This instrument, validated to assess moderate–vigorous PA (MVPA), provides estimates of MVPA and SB that approximated values from an objective monitor [26,27]. This self-report questionnaire was designed to discover the PA reached in the last 7 days and was designed for use in children and adolescents from 8 to 17 years old (grades 4–12). It contained 3 items on activities and sports performed during the last week. The items were: (1) activity at school, (2) activity out of school, and (3) sedentary behaviours. The first item includes PA during breaks, classes, and lunchtime. The second item is PA outside school, including PA before and after school and on the weekends. The last item contains questions about SB, such as time video games, computer time, television time, and telephone time, excluding time spent studying. The items were rated on a Likert-type scale from 1 to 5 (1 = low PA; 5 = high PA). 

A translation of the original questionnaire into the Spanish language was conducted; two independent Spanish researchers with English knowledge translated the original YAP into Spanish. Then, differences were adjusted to reach a consensus. Second, the Spanish version was back-translated to English by two other independent researchers. Finally, a different researcher fluent in English compared the original YAP with the new version translated into English. The new version was called YAP-S (Youth Activity Profile in Spanish). A pilot administration was conducted of the YAP-S in a small group of children and adolescents (n = 20) and additional refinements were made based on the feedback [27].

The YAP score was transformed using Fairclough’s equations [27] to minutes per day of MVPA. To establish whether the subjects were physically active, time > 60 min MVPA/day was considered, and those who did not comply were considered physically inactive (<60 min/day). To determine active travel, the PACO: pedal and walk to school questionnaire, a self-report of travel mode, distance, and travel time to and from school was used. A study of reliability was performed through the kappa coefficient, weighted kappa, and intraclass correlation coefficient (ICC), and its respective confidence interval (CI) for questions about active commuting in children and adolescents in Chile showed high reliability [28,29]. The instrument used to determine self-perceived physical fitness in schoolchildren was the International Fitness Scale (IFIS) questionnaire, which corresponds to a validated test for the assessment of general physical fitness [23], and has 5 items corresponding to the components of general physical fitness, cardiorespiratory fitness, muscular strength, speed/agility, and flexibility. The response section gives options by a Likert-type scale from 1 to 5 (1 = low PA; 5 = high PA) in each question before and during the pandemic. This test was chosen because the restrictions of the second period of the pandemic made it impossible to evaluate fitness objectively or in a more practical context. However, this section from the questionnaire was not used for the analysis in this present report.

### 2.3. Procedures

School principals were invited from the cities of Viña del Mar and Concón (Valparaíso region) and Talcahuano and Concepción (Biobío region) in Chile. Of the schools that agreed to participate, the parents were informed of the objectives and characteristics of the study. The instrument in the Biobío region was applied in an online format because they were in quarantine. In the Valparaíso region, it was applied in paper format because the schools were in hybrid and face-to-face classes. In the online format, the student’s parents or caregivers received the questionnaire through a link via email and WhatsApp (Mountain View, CA, USA). The online questionnaire was conducted through the SurveyMonkey platform (San Mateo, CA, USA). In paper format, the questionnaire was administered to the children in person in the classroom with the help of the physical education teacher. Parents or caregivers were sent the questionnaire home in a sealed envelope and had a maximum of 7 days to return it with their respective answers. For both versions, the questionnaire was applied in August and September 2021, and its application lasted approximately 14 min. 

### 2.4. Ethical Aspects

Prior to completing the questionnaire, the parents or guardians responsible for the schoolchildren received informed consent, where they were notified of the characteristics of the questionnaire and the objectives of the study. After reading and accepting the consent, the children agreed to participate in the research in a document specifying the characteristics of the study and its objective. This research was approved by the ethical committee of the Pontificia Universidad Católica de Valparaíso (BIOEPUCV-H 363-2020).

### 2.5. Statistical Analysis

Statistical analysis was performed using SPSS version 25. Continuous values, such as the level of physical activity and sedentary behaviour, are presented as mean and standard deviation (M ± SD). Categorical variables are presented as frequencies (%). To compare the sociodemographic factors, the Chi-square test was used. For the comparison of means of continuous variables, Student’s T-test was performed. GraphPad Prism (GraphPad, San Diego, CA, USA) was used for graph design. The level of statistical significance was set at *p* < 0.05, with 95% confidence.

## 3. Results

A total of 315 schoolchildren (26% boys and 74% girls) and their respective parents belong to the public (37.8%) and private (62.2%) schools. Of the total families, some had only one child (25.4%, of which 32.5% were boys and 22.4% were girls) while others had more than one child (74.6%, of which 67.5% were boys and 77.6% were girls). The availability of a car was also considered, and we separated those families that did not have a car (24.1%) and those that had more than one child (74.6%). Finally, we also separated the families by socioeconomic level: low (39.8%), medium (28.7%), and high (31.5%).

Table 1 shows the sociodemographic characteristics of the schoolchildren in the study. Within these, significant differences were found in the variables of socioeconomic level, such as age (higher in girls than boys) and type of school, where being enrolled in private school had a higher prevalence among girls. Additionally, car availability was higher in girls than boys, and, in consequence, socioeconomic status was lower in boys than girls. However, the number of children by sex was not significant.

Table 2 shows the SB before and during the pandemic. In all SB variables, there were statistical differences by sex. The time spent playing video games was higher in boys than in girls before (*p* < 0.001) and during the pandemic (*p* < 0.001). Similar results were found in computer time, which was higher in boys than in girls both before (*p* < 0.001) and during (*p* < 0.001) the pandemic. Finally, the same thing happened in telephone and television time, which was higher in boys than in girls in both periods. Although this table is not shown, some SSB tended to decrease during the pandemic. These results will be presented more clearly in Table 3.

According to the results, PA compliance remained low (does not meet recommendations) both before and during the pandemic (74.3% and 74.2%, respectively).

Figure 1 shows the comparison between before and during the pandemic of the SB variables. As shown in the figure, all sedentary behaviours, except for time spent playing video games in boys (*p* = 0.618), were significant. These results prove that both before and during the pandemic, the time spent in sedentary activities exceeded > 1 h per day of use. In another way, Figure 2 shows that the PA was low in both sexes, before and during the pandemic.

Table 3 shows the comparison between before and during the pandemic of SB and PA. All comparisons of both SB and PA gave significant differences in both sexes. The telephone time and television time scores increased during the pandemic in boys and girls. However, video game time and computer time scores decreased during the pandemic in both boys and girls. The same was true for min/day of physical activity, which increased significantly during the pandemic in boys (44.3 to 53.9 min/day) and girls (50.9 to 53.2 min/day).

## 4. Discussion

The aim of this study was to compare the levels of SB and PA in relation to sex, as a sociodemographic variable of Chilean schoolchildren before and during the COVID-19 pandemic. 

PA, like physical condition, has been associated with several positive health outcomes [24,25]. PA has been associated with less premature death and is considered an effective primary and secondary preventive strategy for at least 25 chronic medical conditions [24]. Moreover, sedentary time has been associated with multiple detrimental health outcomes, including all-cause mortality, cardiovascular disease, type 2 diabetes, and metabolic syndrome, and could be an independent risk factor for PA [30]. These factors have a series of implications for public health, where many times the focus is more on PA than on sedentary time and behaviours.

The main results show that SB before the pandemic had significant differences when compared by sex, except for time spent on television. During the pandemic, there was no significant difference in television time and telephone time. There were no significant differences by sex before and during the pandemic. All values were significant when comparing the SB and PA scores before and during the pandemic.

In this study, it was found that television time increased significantly between before and during the pandemic. However, when comparing the results by sex, the difference was not significant. Previous research found that children in the early stages of the pandemic increased their screen time on both television and video games by more than 3 h per day [31]. This may have been due to limited mobility and little activity during confinement. In this regard, Ranjbar et al. [32] note that one of the favourite activities during the pandemic while schools were kept closed was watching television (13.8%). Another cause that can be associated with the increase in television time is parental mental health problems, where it was found that the greater the parental anxiety, the more time children spent in sedentary behaviours such as watching television [33].

In addition, this study showed that computer time for more than one hour per day decreased significantly in both boys and girls compared to before and during the pandemic. When comparing the results by sex, the difference is also significant. Previous research found data showing an increase in internet addiction by 36.7% [34] during the pandemic. This could explain the decrease in computer time due to increased time spent on mobile devices and television. On the other hand, other research associated computer time with a worse classroom climate [14]. 

Regarding the time spent playing video games before and during the pandemic in children, although there were changes, these were not significant. We can contrast this information with that provided by Kim and Lee [35], where internet gamers were classified into four profiles: casual, moderate, potential risk, and addicted. Of these profiles, the only ones that had a significant increase were those with a profile addicted to video games. Taking this information into account, we can assume that the time spent playing video games did not have a significant change. This could occur because, in Chile, there are still no cases of video game addiction, possibly because this trend has not become as massive as it is in South Korea or China, where these cases are beginning to be more and more frequent [36].

In another sense, the time of telephone use in boys remained the same before and during the pandemic, but in girls, it increased significantly, by more than one hour. This is consistent with the findings of Cívico et al. [37], where the results show that boys had more problematic use than girls before COVID-19 confinement, according to the perceptions of their families. 

There are records indicating that the amount of time that children and adolescents occupy the telephone is around 2 h per day, which, upon reaching adolescence, increases according to age, reaching an average of 4 h per day and/or 30 h per week [38,39].

However, the time of telephone use in this study has been considered as recreational use; therefore, the increase in its use to perform school activities could be a factor to consider in this increase. 

Regarding compliance with PA recommendations, a slight increase was observed during the pandemic in boys, but this was only maintained in girls. Comparing the average minutes of PA per day, both girls and boys increased the time.

According to the study conducted by Rossi et al. [12], PA increased as a result of factors that allowed children and adolescents to be more physically active, among the most important of which is being male, since it would benefit the increase in minutes of PA. Other important factors were complying with a daily routine, performing PA outdoors, and having more free time by not attending school in person, among others. 

However, the mean time spent by both groups is below compliance with the PA recommendations (60 min of moderate–vigorous physical activity). A possible explanation for this could be that, according to previous studies, there is a worldwide prevalence of physical inactivity. According to the scientific study ANIBES [40], 55.4% of children and adolescents do not comply with the physical activity recommendations proposed by the World Health Organization (>60 min MVPA/day). 

Based on the results obtained, future research could collect data to determine which sociodemographic factor most affects the prevalence of compliance with PA recommendations. In addition, the study could be replicated in other regions of the country to compare the results and obtain a representative sample of the situation at the national level. In addition, the situation could be compared with developed countries and the causes of the possible results could be investigated. Having all these data will allow the design of action plans to intervene in the population and seek to correct the problems in complying with PA recommendations.

## 5. Strengths and Limitations

The main limitation has been the collection of data from an online questionnaire, with the subjectivity that the use of these instruments implies. However, it was not possible to incorporate more objective methods, due to the situation of intermittent confinement and the restrictions decreed during the pandemic.

As a strength, the incorporation of an adequate sample, from various regions of the country that could be extrapolated to other realities, given the similarity of the pandemic situation, stands out. In addition, a complete variety of sedentary behaviours was incorporated, which allows for a more complete vision of the situation.

## 6. Conclusions

Sedentary behaviour such as time on the telephone and television increased during the pandemic in the whole sample of Chilean schoolchildren, while time on video games and computers decreased. The comparison with Student’s T-test between sexes in SB, such as television time, telephone time, and computer time, was statistically significant (*p* < 0.001). From this research, it was possible to determine that the mean PA increased in boys (Δ + 9.51 min) and girls (Δ + 3.54 min) during the pandemic (*p* < 0.001), since people’s interest in participating in sports increased during the pandemic due to the need to move after months of confinement. Despite the increase in PA time, schoolchildren failed to meet the PA recommendations established by the World Health Organization (>60 min MVPA/day). 

As for the practical implications of this work, strategies should be implemented at school to promote and increase PA practice in this current period, which could be called “post-pandemic”. In addition, effective government proposals to increase PA are needed to satisfy the already existing interest in the population. 

Further research is needed to determine whether other sociodemographic factors may have influenced the PA outcomes of Chilean schoolchildren and compliance with the recommendations. Second, the future research should investigate the medium and long-term effects of the pandemic on PA and SB. 

Finally, many lessons were learned from the pandemic in various areas, including PA and SB. The system was not prepared to deal with the health situation with a certain normality, and caused harmful effects on people’s PA. In the future, the use of technology must be in favour, and physical practice in open spaces cannot be prohibited. Rather, reforms should be made in order to provide the multiple and powerful social, mental, and health benefits of PA.

## Figures and Tables

**Figure 1 children-10-00481-f001:**
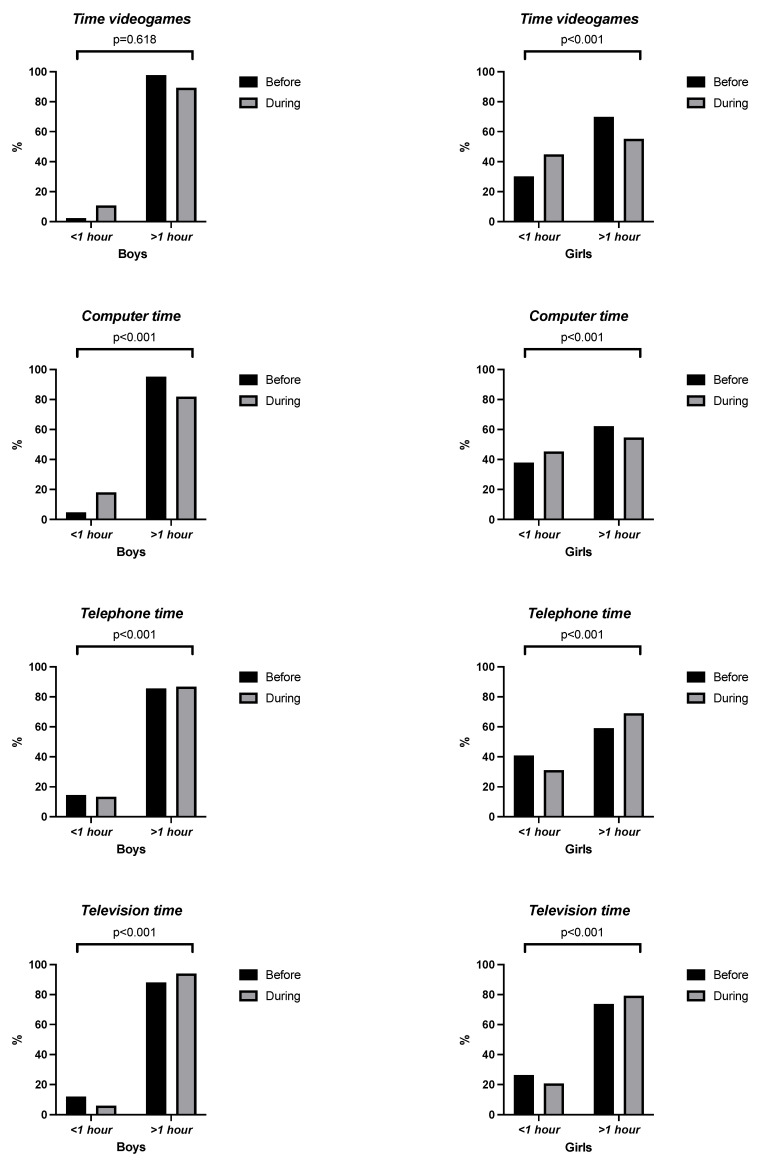
Comparison of sedentary behaviours before and during the pandemic by time category.

**Figure 2 children-10-00481-f002:**
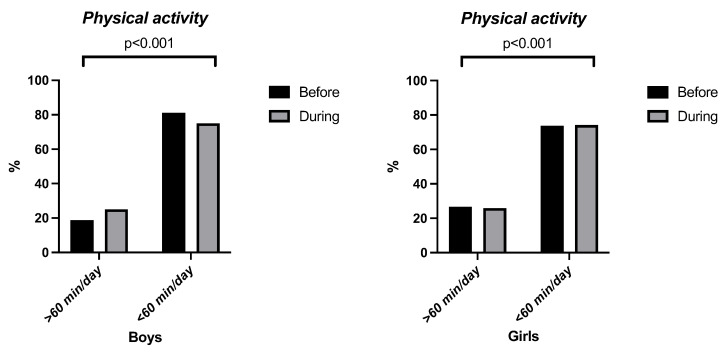
Comparison of physical activity before and during the pandemic by time category.

**Table 1 children-10-00481-t001:** Sociodemographic factors and characteristics of the schoolchildren.

Sociodemographic Factors	All	Boys	Girls	*p*-Value
	(n = 315)	(n = 83)	(n = 232)	
	N (%)	N (%)	N (%)	
Age (Mean ± SD)	10.9 ± 1.9	9.3 ± 2.0	11.0 ± 1.8	*p* < 0.001
Children per family				
Only one	72 (25.4)	27 (32.5)	45 (22.4)	*p* = 0.074
More than one	212 (74.6)	56 (67.5)	156 (77.6)	
School type				
Private	196 (62.2)	38 (45.8)	158 (68.1)	*p* < 0.001
Public	119 (37.8)	45 (54.2)	74 (31.9)	
Car availability				
None	34 (24.1)	10 (62.5)	24 (19.2)	*p* < 0.001
Only one	29 (20.6)	6 (37.5)	23 (18.4)	
More than one	78 (55.3)	0 (0.0)	78 (62.4)	
Socioeconomic status				
Low	43 (39.8)	16 (100)	27 (29.3)	*p* < 0.001
Middle	31 (28.7)	0 (0.0)	31 (33.7)	
High	34 (31.5)	0 (0.0)	34 (37)	

SD: Standard deviation; Statistical differences in *p* < 0.001.

**Table 2 children-10-00481-t002:** Sedentary behaviours in times of rest before and during the pandemic.

	All	Girls	Boys	
	N (%)	N (%)	N (%)	*p*-Value
Sedentary behaviours				
Time video games before the pandemic	315 (100)	232 (73.7)	83 (26.3)	*p* < 0.001
Less than one hour	72 (22.9)	70 (30.2)	2 (2.4)	
More than one hour	243 (77.1)	162 (69.8)	81 (97.6)	
Time video games during the pandemic	315 (100)	232 (73.7)	83 (26.3)	*p* < 0.001
Less than one hour	113 (35.9)	104 (44.8)	9 (10.8)	
More than one hour	202 (64.1)	128 (55.2)	74 (89.2)	
Computer time before the pandemic	315 (100)	232 (73.7)	83 (26.3)	*p* < 0.001
Less than one hour	92 (29.2)	88 (37.9)	4 (4.8)	
More than one hour	223 (70.8)	144 (62.1)	79 (95.2)	
Computer time during the pandemic	315 (100)	232 (73.3)	83 (26.3)	*p* < 0.001
Less than one hour	120 (38.1)	105 (45.3)	15 (18.1)	
More than one hour	195 (61.9)	127 (54.7)	68 (81.9)	
Telephone time before the pandemic	315 (100)	232 (73.7)	83 (26.3)	*p* < 0.001
Less than one hour	107 (34)	95 (40.9)	12 (14.5)	
More than one hour	208 (66)	137 (59.1)	71 (85.5)	
Telephone time during the pandemic	315 (100)	232 (73.7)	83 (26.3)	*p* = 0.002
Less than one hour	83 (26.3)	72 (31.0)	11 (13.3)	
More than one hour	232 (73.7)	160 (69)	72 (86.7)	
Television time before the pandemic	315 (100)	232 (73.7)	83 (26.3)	*p* = 0.008
Less than one hour	71 (22.5)	61 (26.3)	10 (12)	
More than one hour	244 (77.5)	171 (73.7)	73 (88)	
Television time during the pandemic	315 (100)	232 (73.7)	83 (26.3)	*p* = 0.002
Less than one hour	53 (16.8)	48 (20.7)	5 (6.0)	
More than one hour	262 (83.2)	184 (79.3)	78 (94.0)	
Physical activity before the pandemic				
Complies	38 (35.7)	35 (26.5)	3 (18.8)	*p* = 0.502
Does not meet	110 (74.3)	97 (73.5)	13 (81.3)	
Physical activity during the pandemic				
Complies	40 (25.8)	36 (25.9)	4 (25)	*p* = 0.938
Does not meet	115 (74.2)	103 (74.1)	12 (75)	

**Table 3 children-10-00481-t003:** Comparison of sedentary behaviours and PA before and during the pandemic.

	Boys			Girls		
	Before	During		Before	During	
	M (SD)	M (SD)	*p*-Value	M (SD)	M (SD)	*p*-Value
Score Video games time	3.6 (1.1)	2.4 (1.0)	*p* < 0.001	2.7 (1.3)	1.9 (1.0)	*p* < 0.001
Score Computer time	3.3 (1.3)	1.3 (0.6)	*p* < 0.001	2.3 (1.32)	1.8 (1.0)	*p* < 0.001
Score Telephone time	1.9 (1.5)	2.1 (1.1)	*p* < 0.001	2.2 (1.2)	2.5 (1.2)	*p* < 0.001
Score Television time	2.6 (1.3)	3.0 (1.1)	*p* < 0.001	2.7 (1.1)	3.1 (1.0)	*p* < 0.001
Physical activity (Min/day)	44.3 (21.9)	53.9 (21.5)	*p* < 0.001	50.9 (15.9)	53.2 (19.5)	*p* < 0.001

## Data Availability

In this link https://drive.google.com/drive/folders/1y1mDk4W6txdeZ_xF9MQ4Ih_g-eyhaqp1?usp=share_link.

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
