# Peer review of "Sedentary Behaviour and Physical Activity Levels during Second Period of Lockdown in Chilean’s Schoolchildren: How Bad Is It?"

_children, 2023, doi:10.3390/children10030481_

Round 1

Reviewer 1 Report

1.Revise your introduction section as "Background, Sedentary behaviour and physical activity level during second 2 period of lockdown at Global Level, Sedentary behaviour and physical activity level during second 2 period of lockdown in  Chilean’s schoolchildren, Research gap, Statement of the study including objectives

2. Methodology; Provide your data collection tool as a supplementary file

3. Rewrite the conclusion section again including implications of the study results and future scope

Author Response

Thank you for your review. It helps us to improve.

Reviewer 2 Report

Congratulations for the work, but the research results are predictable in the context of the restrictions caused by the pandemic.

Introduction: The introduction is clear and refers to the results obtained;

Citations: The citations support the context of the research, but it must be improved with articles that already talk about the predictability of its specificity;

Methodology of research 

There are some deficiencies determined by:

1. the lack of the Alpha Cronbach coefficient for the questionnaire (which shows the degree of fidelity of the items);

2. the groups are not equivalent, regarding the listed variables (gender, type of institution, etc.);

3. r (Pearson) or p (Spearman) values are not interpreted (if a relationship between PA and SB is to be identified)

4. the values of r (Pearson) or p (Spearman) are not interpreted (if one wants to identify a relationship between PA and SB) the value;

5. what is the result of linear regression?

Conclusions: the conclusion regarding the statistical methods mentioned in the content of the work is missing

Author Response

Thank you for your comments. It helps us to improve.

Round 2

Reviewer 1 Report

NA

Author Response

Thank you.

Reviewer 2 Report

Thanks to the authors for the corrections, but the final decision is the one from the first evaluation.

Author Response

Thank you.